# Aerosol effects on clouds are concealed by natural cloud heterogeneity and satellite retrieval errors

Antti Arola ⓘ[1] ✉, Antti Lipponen ⓘ[1], Pekka Kolmonen[1], Timo H. Virtanen ⓘ[1], Nicolas Bellouin ⓘ[2], Daniel P. Grosvenor[3], Edward Gryspeerdt ⓘ[4], Johannes Quaas ⓘ[5] & Harri Kokkola ⓘ[1]

One major source of uncertainty in the cloud-mediated aerosol forcing arises from the magnitude of the cloud liquid water path (LWP) adjustment to aerosol-cloud interactions, which is poorly constrained by observations. Many of the recent satellite-based studies have observed a decreasing LWP as a function of cloud droplet number concentration (CDNC) as the dominating behavior. Estimating the LWP response to the CDNC changes is a complex task since various confounding factors need to be isolated. However, an important aspect has not been sufficiently considered: the propagation of natural spatial variability and errors in satellite retrievals of cloud optical depth and cloud effective radius to estimates of CDNC and LWP. Here we use satellite and simulated measurements to demonstrate that, because of this propagation, even a positive LWP adjustment is likely to be misinterpreted as negative. This biasing effect therefore leads to an underestimate of the aerosol-cloud-climate cooling and must be properly considered in future studies.

The climate warming effect of increases in greenhouse gases (GHGs) has been offset to some extent by a cooling effect induced by emissions of anthropogenic aerosols and their precursors[1]. Aerosols affect the Earth's energy budget and climate directly by scattering and absorption of radiation and indirectly by the alteration of cloud properties via increasing the number of cloud condensation nuclei (CCN) and the cloud droplet number concentration (CDNC). The total effective radiative forcing due to this increase in aerosols is the most uncertain component of the historical radiative forcing of Earth's climate[1]. The most significant part of this aerosol related uncertainty is linked to aerosol-cloud interactions (ACI)[1,2].

The effect of ACI, expressed as an aerosol-induced perturbation of the net radiative energy flux into the climate system, is typically quantified as the effective radiative forcing of ACI ($ERF_{aci}$). $ERF_{aci}$ from liquid water clouds probably dominates the total $ERF_{aci}$[2] and is usually divided into two components: (1) an instantaneous radiative forcing induced by an increase in CDNC, often called the Twomey effect[3], the

cloud albedo effect, or the 1st indirect effect, and (2) rapid adjustments of other cloud properties in response to this increase in CDNC. The most important pathways of these rapid adjustments are the CDNC driven changes in the cloud liquid water path (LWP) and in the cloud fraction (CF), with corresponding forcings.

It has been hypothesized that an increase in aerosol load can cause an increase in the cloud liquid water content through a delay in precipitation[4] that would manifest itself as a positive cloud LWP adjustment. On the other hand, modeling evidence has shown that the altered droplet size distributions can affect the entrainment mixing of clouds with the surrounding dryer air thus reducing LWP (e.g., ref. 5). Early studies investigating the LWP response to aerosol perturbations[6-9] considered the relationship of LWP with aerosol optical depth or related quantities. Such aerosol measurements are prone to aerosol swelling effects leading to flawed relationships[10] between LWP and aerosols. Also, as demonstrated by Ma et al.[11], the low aerosol loading conditions are typically not well characterized by

[1]Finnish Meteorological Institute, Kuopio, Finland. [2]Department of Meteorology, University of Reading, Reading, UK. [3]School of Earth and Environment, University of Leeds, Leeds, UK. [4]Space and Atmospheric Physics Group, Imperial College London, London, UK. [5]Institute for Meteorology, Leipzig University, Leipzig, Germany. ✉e-mail: antti.arola@fmi.fi

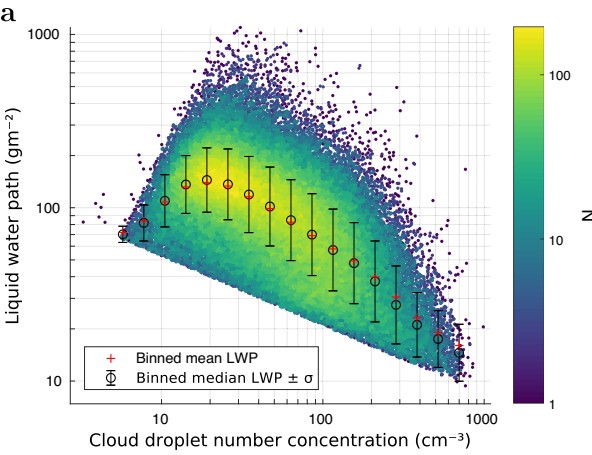

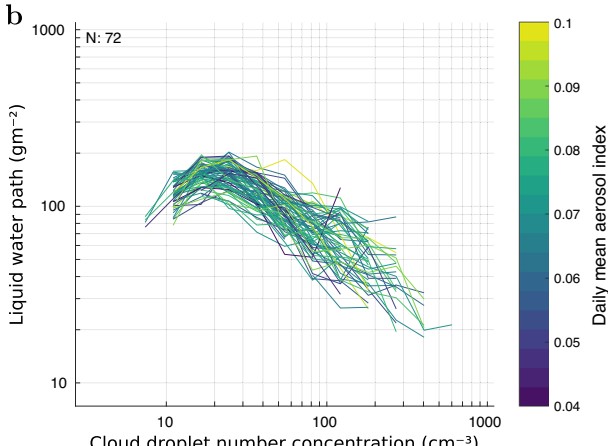

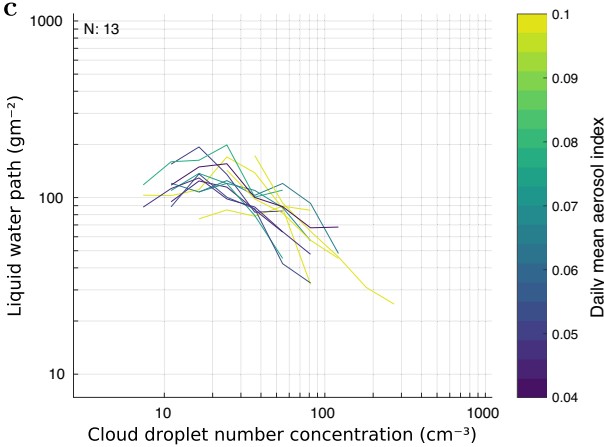

**Fig. 1 | Liquid water path (LWP) vs cloud droplet number concentration (CDNC) relationships obtained from satellite data over decreasing spatiotemporal scales.** Similar increasing and then decreasing paired LWP vs CDNC relationships are observed in all cases indicating that processes occurring at small spatio-temporal scales are causing the relationships. Panel **a** shows a 2D histogram of LWP and CDNC 1 × 1° daily datapoints taken over 1825 days for the whole of the Pacific North region (20–35° N, 110–150° W; ocean only) with colors showing the number of datapoints and the circles/bars showing the mean LWP ± the standard deviation for specific CDNC bins. Panel **b** shows LWP binned by CDNC for the same area and period, but now using only daily snapshot data for each line in order to limit meteorological variability over time. Additionally, the daily standard deviation of the Aerosol Index is limited to below 0.04 in order to limit the impact of aerosol changes on the observed relationships. N shows the number of days in the plot and only those days are shown for which the number of datapoints $N_{day}$ exceeds 150. Only those points are shown for which the number of datapoints per CDNC bin exceeds 4. Lines are colored by the area mean AI. Panel **c** shows the same as for **b** except that a much smaller region (5 × 5°) is used in order to limit meteorological variability, and thresholds $N_{day} > 100$ and NB > 3 are used. MODIS data aggregated to 0.25 × 0.25° scale was used in **c** to increase the number of datapoints.

sufficiently considered before: the treatment of retrieval errors in the satellite-based cloud optical depth (COD) and cloud effective radius (CER) retrievals, and the effects due to spatial heterogeneity in cloud fields caused by, for example, clouds in different stages of their development and spatial variability in updraft velocities. There are several sources of uncertainty in cloud retrievals, such as the sub-pixel inhomogeneity or 3-D cloud effects. These and other sources of uncertainty are discussed in detail in e.g., Grosvenor et al.[17]. However, based on the cloud fields we examined more closely in this study, it would seem that the natural heterogeneity of cloud fields plays a more substantial role in causing a bias in the LWP adjustment estimate than COD or CER retrieval uncertainties. The errors and spatial hetero-geneity in COD and CER retrievals propagate into CDNC and LWP[17] and are very likely to confuse the interpretation of LWP adjustments using the commonly-employed method of logarithmic-scale relationships between LWP and CDNC[10,15]. In this paper, we first show how errors in real satellite COD and CER retrievals or spatial heterogeneity in cloud fields influence the CDNC-LWP relation. Then, we use synthetic data to demonstrate how adding realistic retrieval errors and spatial variability changes an imposed strongly positive CDNC-LWP relationship into a negative one of the type seen in previous studies (e.g., refs. 10, 15).

## Results

### CDNC-LWP patterns obtained with MODIS aerosol and cloud products

We analyzed level-2 (L2) Moderate Resolution Imaging Spectro-radiometer (MODIS) data over the ocean, both for aerosol and cloud products, covering four subtropical regions that are often focused on in satellite-based studies of aerosol-cloud interactions, namely the North and South Atlantic and the North and South Pacific. We calcu-lated CDNC and LWP using 1 km L2 resolution cloud products, as explained in more detail in "Methods", and then aggregated them into 0.25° or 1° spatial resolutions depending on the particular analysis being performed.

First we examine the relationships obtained over large spatial regions of order a few thousand kilometers across and using data gathered over a period of five years (2014–2018). Figure 1a shows the results for the Pacific North with the other regions shown in Supple-mentary Figs. 1–3. In all four regions we observed similar logarithmic-scale relationships between LWP and CDNC to those that have been shown previously in the literature (e.g., in Fig. 3 in Gryspeerdt et al.[10] or in Fig. 4 in Possner et al.[15]). Over such large spatial and temporal scales it is likely that co-variability between aerosols and clouds will be influencing the relationship between LWP and CDNC. For example, particular climatological meteorological features in one part of a

satellites, conditions that are the most relevant ones for the aerosol-cloud interaction studies. Only rather recently have there been satellite-based studies on LWP adjustments that use CDNC as an aerosol proxy. The results have not given a clear and consistent picture about the LWP adjustment effect; some studies have postulated that changes in CDNC can enhance LWP[12], some showed that they have only a minimal overall effect[13], and others have shown that enhanced CDNC can even reduce LWP[10,14,15]. It is a complex task to estimate the LWP responses to the CDNC changes since various confounding factors need to be isolated, e.g., the co-variability between CDNC and LWP induced by meteorological effects[15]. As such, it is challenging to isolate and estimate the true one-way causal effect[16].

We argue here that there are important aspects of the estimation of LWP adjustments using satellite observations that have not been

region may be associated with low CDNC and high LWP whereas the meteorological regime in another part of the region may be associated with high CDNC and low LWP; combining the two would produce a negative relationship between LWP and CDNC.

To combat this we also consider the relationships obtained when considering only one snapshot in time (Fig. 1b), which removes the effect of meteorological variability over time and then further restrict the analysis to much smaller regions (5 × 5°; Fig. 1c) in order to reduce the impact of larger scale spatial meteorological variability. Furthermore, we only show the LWP vs CDNC relationships from those snapshots where the spatial aerosol variability was very low in order to choose cloud fields with a constant aerosol load. In such cases the dependence of relationships between LWP and CDNC on confounding large-scale meteorological factors or changes in CCN concentrations should be reduced. The relationships would therefore be most likely due to satellite retrieval errors in the COD and CER variables used to estimate CDNC and LWP, or to be due to natural spatial heterogeneity in COD and CER fields occurring in such a way as to violate the assumptions made for the CDNC and LWP retrievals (see Methods).

As can be concluded from both Fig. 1b, c (and from the Supplementary Figs. 1–3), the patterns are very similar no matter the spatial scale: a negative CDNC vs. LWP slope appears at higher cloud droplet concentrations. Since we deliberately excluded the effects of aerosols and large-scale meteorology here, the CDNC-LWP patterns are mainly due to spatial heterogeneity in the cloud fields or retrieval errors in LWP or CDNC. Spatial variability could arise due to variation in updraft velocities, varying stages of cloud development, varying degrees of cloud top entrainment, etc., which may introduce changes in CER or COD that are not consistent with the assumptions of the CDNC and LWP retrievals (see Methods) and may therefore cause biases in the CDNC and LWP values. CER and COD retrieval errors would also introduce biases in CDNC and LWP. Hence the obtained relationship between them could also be biased.

The LWP vs CDNC relationships can be also affected by the occurrence of precipitation (e.g., refs. 16, 18). Therefore, we made one additional analysis by separating our dataset to include only cases when CER was smaller than 15 μm, which threshold has been often used as an approximate indicator of precipitation (e.g., refs. 19, 10). These results are shown in Supplementary Fig. 5. Regardless of how we separately focus on raining vs. non-raining clouds, it is apparent that the biasing effect exists and becomes even more obvious in the conditions of non-raining clouds. This is fully understandable and in line with our main message: the spatial variability of CER introduces a bias which moreover becomes stronger in conditions where the CER values are lower on average.

## On CDNC-LWP patterns obtained with simulated measurements

Now we use simulated satellite measurements with varying levels of spatial variability and retrieval error in COD and CER to show how they can lead to the misinterpretation of the LWP-CDNC relationship. We could not separate these two effects in our simulations and so we are, strictly speaking, simulating the impact of both spatial variability and retrieval errors, which will be referred to collectively as "error" from here onwards.

We constructed a simulated dataset by assuming adiabatic cloud liquid water content profiles with constant CDNC throughout the cloud depth as is also assumed for CDNC and LWP satellite retrievals. Starting with given LWP and CDNC values we varied the CDNC according to a prescribed dlnCDNC/dlnCCN value, and also assumed a prescribed LWP adjustment effect (described in detail in "Methods"). This dataset forms our "truth" cloud dataset with a built-in positive relationship between LWP and CDNC. We then also created a dataset in which we separately applied CER and COD errors to the values from the truth dataset using samples from a normal distribution of errors with zero mean and with a given standard deviation ($\sigma_e$). These were

then used to generate a new set of LWP and CDNC values to simulate the effect of random errors and variability.

Figure 2a shows one simulated example case of COD and CER with errors. In this case we set a lower bound for CDNC of 40 cm$^{-3}$ corresponding to a CCN value of 100 cm$^{-3}$ and LWP of 80 g/m$^2$. CDNC was set to increase towards higher CCN amounts following a fixed dlnCDNC/dlnCCN value of 0.8. The LWP adjustment was set as dlnLWP/dlnCDNC = 0.5. With these selections, our overall range in LWP was similar to that observed in some previous studies (e.g., ref. 10). The LWP adjustment that we assume is stronger than that found in earlier studies. In this example, we deliberately introduced a strong positive LWP adjustment to demonstrate that the data with errors can easily disguise a positive adjustment and lead to a negative or ambiguous diagnosed LWP-to-CDNC relationship. In Fig. 2a the simulated CDNC is shown against LWP. The true imposed LWP adjustment in the error-free dataset is shown by circles. Despite the assumption of a positive LWP response to CDNC, it seems unlikely from the form of the data cloud that the prescribed positive slope could be obtained by fitting a linear regression line using standard methods.

From the Fig. 2b where the data has been binned by CDNC it becomes clear that this emerging LWP adjustment pattern very much resembles the increasing and then decreasing LWP vs CDNC relationship obtained in several earlier studies[10,15] and in the previous section. In this case, a relative error distribution with $\sigma_e$ = 15% in COD and CER was used, while the Fig. 2c shows the effect of varying the relative and absolute error. As the errors in COD and CER increase, the peak in the LWP curve moves towards a lower CDNC (Fig. 2c). In all cases here the prescribed LWP adjustment of dlnLWP/dlnCDNC was a positive value of 0.5, which is illustrated by the simulation with zero error in COD and CER (black line).

In the analysis shown in Fig. 2 we introduced various levels of variability/error into COT and CER, while the Equation (1) suggests that the possible variability of $\alpha$ would introduce a similar effect to COD but more strongly (COD is taken to the power of 0.5). This variability could be caused, for example, by the fact that the true adiabatic fraction deviates from constant sub-adiabacity assumed in the bulk coefficient of $\alpha$. This motivated us to make one additional simulation, where COT and CER uncertainties were 15%, and the relative variability/error of $\alpha$ parameter was 0%, 15%, and 25%. Those results are shown in Supplementary Fig. 4, and show that the descending branch of LWP vs. CDNC starts with somewhat higher CDNC values, if $\alpha$ is varied too. However, the main pattern and the negative "LWP branch" is there, still in line with the main message of our study.

Overwhelmingly, the main cause of the negative LWP vs CDNC slopes is the error in CER. The error in COD causes a slightly more positive slope than that imposed, but it is a very small increase. Figure 3 shows simulations where the impacts of the relative COD or CER error distributions (with $\sigma_e$ = 25%) were separately removed for the CDNC and LWP calculations in order to test their effect on the LWP vs CDNC relationship. The combined errors for both COD and CER were also removed for just the CDNC calculations and then for just the LWP calculations. There are a few details worth highlighting. First, in the case where the COD and CER errors affected only the LWP calculation (panel (d) and purple line in panel (f)), the observed relationship becomes very close to the black line case of no error. This should not be surprising, since the data are binned against CDNC. Then, of course, the relative error in COD and CER causes a significant error in LWP, which however gets almost averaged out and the median LWP for each CDNC bin is very close to the no error case. This suggests that any improvements in data quality should be focused on the CDNC calculation rather than in the LWP calculation; this fact also likely explains why using LWP data from the Advanced Microwave Scanning Radiometer for EOS (AMSR-E) produces LWP vs CDNC relationships that are not substantially different from that obtained using MODIS LWP data. The second point is that it is the variability in CER that is the main

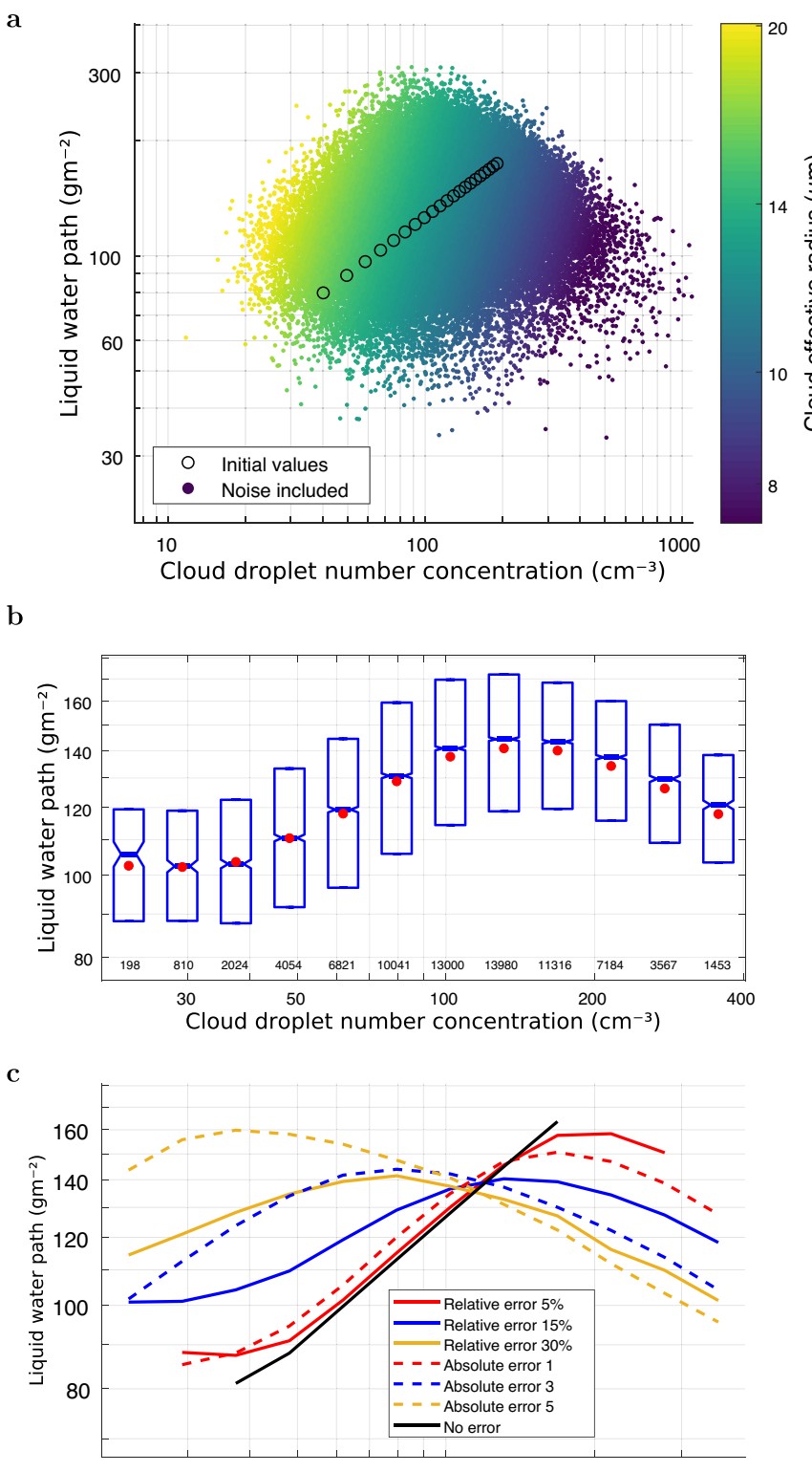

**Fig. 2 | Simulated liquid water path (LWP) against cloud droplet number concentration (CDNC) for the case of cloud optical depth (COD) and cloud effective radius (CER) error of 15%.** dlnCDNC/dlnCCN = 0.8 and dlnLWP/dlnCDNC = 0.5 were prescribed. The initial values of 100 cm⁻³ for cloud condensation nuclei (CCN), 40 cm⁻³ for CDNC and 80 gm⁻² for LWP were assumed. **a** The CER is shown as color bar. The circles show the imposed relationship in the LWP adjustment (dlnLWP/dlnCDNC). **b** The box-plot of the same results shown in panel **a**. The red dots show the mean, blue horizontal lines show the median and the box height is determined by the first and third quartiles. **c** Median values resulting from varying levels of relative/absolute error. Unit of absolute CER error is μm and absolute COD error is unitless.

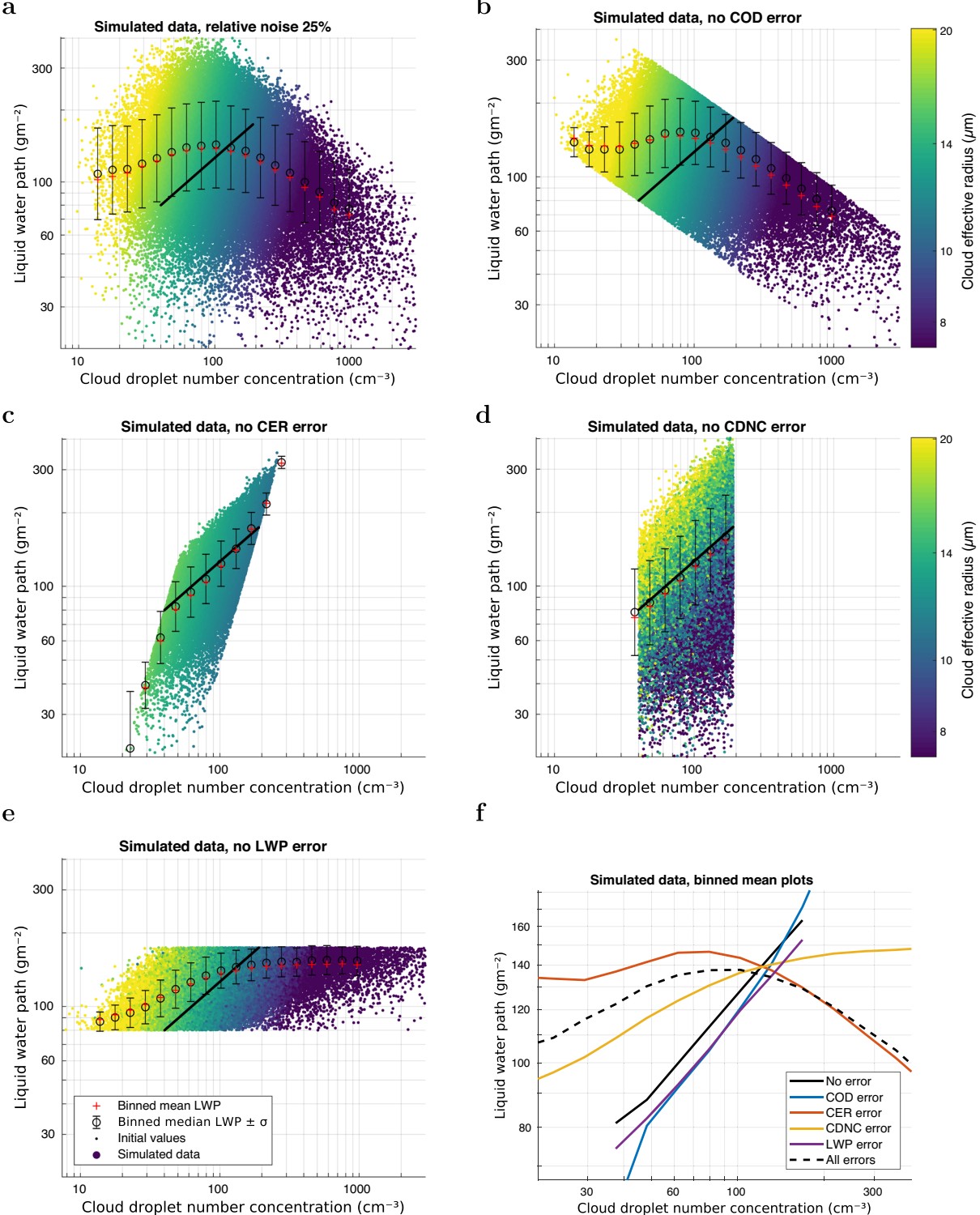

**Fig. 3 | Cloud droplet number concentration (CDNC) vs liquid water path (LWP) patterns in different cases of cloud optical depth (COD) and/or cloud effective radius (CER) variability/error.** Relative error of 25% was used, and LWP adjustment was set to 0.5. **a** Variability was applied to both CER and COD and propagated to CDNC and LWP. **b** Variability was not applied to COD, only to CER. **c** Variability was not applied to CER, only to COD. **d** Variability was applied to both COD and CER, but was not propagated to CDNC, only to LWP. **e** Variability was not propagated to LWP, only to CDNC. **f** Combined plot of the binned mean values from cases **a**–**e**, and the initial values without any variability (No errors, solid black line). Both COD and CER were without variability (black line); CER was accurate but COD included variability/error (blue line); COD was accurate but CER included variability (red line); CER and COD errors influenced only LWP calculation (purple line); CER and COD errors influenced only CDNC calculation (yellow line); CER and COD errors influenced both LWP and CDNC calculation (dashed black line).

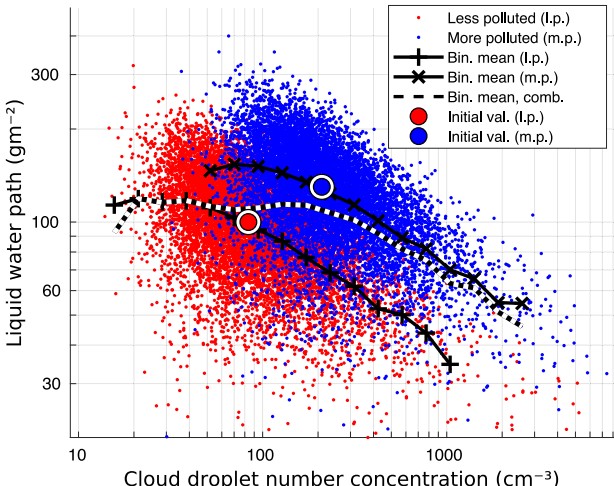

**Fig. 4 | Simulation of less and more polluted cases.** For less polluted case (red dots) assuming values of cloud effective radius (CER) and liquid water path (LWP) of 13 μm and 100 gm⁻², respectively, and for more polluted case (blue dots) values of CER and LWP of 10 μm and 130 gm⁻², respectively. Relative error of 25% was assumed for both cloud optical depth (COD) and CER. Cloud droplet number concentration (CDNC) binned mean values of LWP are shown for both cases separately and also for the combined dataset including both cases (dashed line). Large red and blue circles show the initial values for CDNC and LWP for the less and more polluted case, respectively.

reason for the negative slope, which can be seen from the red line case where only the error in CER was included.

In the previous simulation case, the CCN concentration increased from 100 to 700 cm⁻³, which corresponded to CDNC values from 40 to 190 cm⁻³ given our assumed relationship between CDNC and CCN. But perhaps even more visually clear is to consider only two separate cases of CCN concentrations, representing for example less and more polluted sectors; e.g., out-of-plume and in-plume regions in analyses of aerosol plumes or pre-industrial and present-day conditions. Therefore, we also simulated a less polluted case assuming values of CER and LWP of 13 μm and 100 gm⁻², respectively, and a more polluted case values of CER and LWP of 10 μm and 130 gm⁻², respectively. These values were assessed based the Table in Toll et al.[14], as typical/reasonable values to represent these two sectors. We then applied a relative error distribution with $\sigma_e = 25\%$ for both COD and CER. In Fig. 4 CDNC-binned mean values of LWP are shown separately for both cases and for the combined dataset (dashed line). The prescribed conditions are shown by large circles, whereas the variation in individual "observations" due to the imposed error distributions, is considerable and bends both the less and more polluted LWP lines down as a function of the CDNC. The dotted line shows the mean LWP of the combined dataset and thus represents the pattern that has been often interpreted as LWP adjustment. However, it is to be emphasized that the errors in both datasets are only due to the variability in COD and CER we introduced and only the comparison between the low and high CCN datasets could reveal the cloud response to aerosol perturbation.

In the simulations shown in Fig. 4, the errors in COD and CER were not correlated. In Supplemenary Fig. 6, we show also cases when there was a perfect positive (Supplementary Fig. 6a) or negative correlation (Supplementary Fig. 6b) between COD and CER errors. As can be seen, the negative slope cannot be avoided, even in the case of perfect negative correlation. On the other hand, it gets very pronounced with the positive correlation and evidence suggests[20] that some level of positive correlation is more likely than negative correlation for COD and CER errors, in other words the kind of cases when optically thicker clouds are more likely to have higher cloud top CER.

Supplementary Figure 7 illustrates the inter-dependencies between LWP and CDNC when calculated by satellite-measured COD and CER using Eq. (1) and (2). If one imagines a positive slope of increasing LWP as a function of CDNC, it is apparent that the error in CER introduces errors in CDNC and LWP roughly along a COD isoline. So this further explains the more important role of CER error. One additional point can be raised from this figure. In many cases a COD threshold of 4 has been used to exclude cases of lower COD. As shown by this figure, in some cases it can contribute to amplifying the negative slope of dlnLWP/dlnCDNC when the threshold of LWP values excluded is increasing with decreasing CDNC. In the Figure, there are also a few circles and lines connecting them to further illustrate the simulation cases of the previous figure. The large red and blue circles show the mean conditions in the less and more polluted cases, respectively. Positive LWP adjustment was again assumed, which is the y-axis distance between the red and blue circles. Given the large variability in data (and illustrated by the Fig. 4), it is not straightforward to isolate the impact on the LWP vs CDNC relationship due only to aerosols from the impact of the errors. However, this is fundamentally important to consider in future studies.

We have emphasized that both the spatial cloud variability and retrieval errors in CER and COT are similar sources for negative bias in LWP adjustment. Regarding our simulations, we also emphasized that it was not possible to separately assess these. It is similarly difficult to separate and quantify the roles of the retrieval errors and spatial cloud variability using satellite data. We tried, however, at least on some level. We analyzed November-December 2018 MODIS data from Pacific South region, day-by-day and visualizing LWP and CDNC fields separately and then their common behavior (LWP vs. CDNC). Very typical case is selected and shown in the Supplementary Fig. 8, demonstrating that even if the MODIS COD and CER data is restricted to cases with lower uncertainty, the negative behavior between LWP and CDNC is very evident, albeit the pattern becomes somewhat less steep. With this data over limited spatial and temporal region, wet get further confirmation that both spatial cloud variability and retrieval errors play a role and indeed biasing to the same direction.

## Discussion

In earlier studies resulting in negative LWP adjustments inferred from satellite measurements, physical cloud processes were discussed as the main candidates for these effects, and there are indeed entirely plausible physical mechanisms that could have produced the inferred relationships. However, it should be stressed that now in our simulated dataset only two possible reasons could have caused any LWP vs. CDNC patterns: (1) the prescribed LWP adjustment, and (2) the retrieval errors and spatial variability in COD and, particularly, in CER. In the particular case shown in Fig. 2, the prescribed LWP adjustment was significantly positive. Nevertheless, we see similar patterns to those obtained in many earlier studies that concluded a negative LWP adjustment effect.

As we demonstrated in this study, the retrieval errors in satellite-based cloud parameters and/or spatial variability in cloud properties can cause a clear bias in estimates of LWP adjustment. It would therefore be extremely important to find a way forward, which could be through much more accurate estimates of LWP and CDNC, the latter in particular. One avenue for improving CDNC estimates could include synergy and data fusion of active and passive remote sensing capabilities. This could also help to alleviate the issues caused by natural variability of the type represented by our simulated dataset whereby there is variability in CER that is inconsistent with the assumptions of the retrievals. On the other hand, even then true spatial cloud field variability remains, which may be causing LWP vs CDNC relationships that are not caused by aerosol variability due to meteorological co-variability, which we have not directly tested in our simulated dataset. Therefore, advances should be made to better explore and characterize such variability.

We believe that one promising way forward could be Bayesian inference, where both cloud albedo and LWP adjustment effects are simultaneously estimated and the underlying errors are modeled and taken into account as thoroughly as possible. The use of Bayesian inference is an ongoing study in our group and will be the topic of a future paper. Another way forward would be to directly retrieve CDNC, rather than inferring it from CER and COD. This would at least partially alleviate the problem and allow for a statistical treatment of measurement errors. This avenue, too, is the subject of ongoing research.

## Methods

### Droplet concentration and LWP estimates

It should be emphasized that in the MODIS cloud product, cloud effective radius (CER, $r_e$) and cloud optical depth (COD, $\tau_c$) are the only retrieval products, while liquid water path (LWP) and cloud droplet number concentration (CDNC) are calculated by specific functional relationships from $r_e$ and $\tau_c$, in our case using the following equations:

$$CDNC = \alpha \times \tau_c^{0.5} \times r_e^{-2.5} \tag{1}$$

$$LWP = 5/9 \times r_e \times \tau_c, \tag{2}$$

where $\alpha$ is $1.37 \times 10^{-5}$ m$^{-0.521}$. These equations have been used in several publications e.g., in the Appendix A of Gryspeerdt et al.[10]. For such derivations it assumed that the cloud liquid water content profile is a constant fraction of its adiabatic value, that the CDNC is constant throughout the cloud depth and that CER is proportional to the volume mean radius. Our version (Equation (1)) is making use of the adiabatic assumption[21].

### Generation of simulated measurements

We constructed a simulated dataset in order to demonstrate how natural variability in cloud properties and errors in cloud property satellite retrievals influence the satellite-based estimate of the aerosol impact on LWP. In our simulations, we varied the relative and absolute levels of random errors in COD and CER. It is to be emphasized that such errors could result both from errors in satellite-retrieved cloud parameters and from natural variability in cloud fields.

We first of all created a baseline ("true") dataset based on the same assumptions used for the CDNC derivation described above. We selected initial values of 100 cm$^{-3}$ for CCN, 40 cm$^{-3}$ for CDNC and 80 gm$^{-2}$ for LWP and calculated the corresponding CER and COD using Eq. (1) and (2). Then, we increased CDNC based on dlnCDNC/dlnCCN = 0.8, followed by an increased LWP based on dlnLWP/dlnCDNC = 0.5, which introduced a positive LWP adjustment effect. CCN was then increased step-wise until it reached a value of 700 cm$^{-3}$. In each step, we calculated CDNC, LWP, CER, and COD. These values can be considered to represent "true" values since none of the variables include spatial variability or retrieval errors.

Satellite-based LWP adjustment effects (the patterns of dlnLWP/dlnCDNC) can potentially get biased due to the variability in COD and CER. This variability in COD and CER may come from two sources. First, satellite-retrieved CER and COD include some amount of error. Second, there is "natural" CER and COD variability over large-scale heterogeneous clouds, which is not related to the aerosol concentration, and which may arise, for example, due to cloud top entrainment or other processes causing clouds to deviate from the assumptions made in the CDNC and LWP derivations. We call this natural variability in this paper.

Because of the above-mentioned retrieval errors and natural spatial variability in cloud parameters, we introduced errors into the CER and COD values and used those values to create a new dataset of CDNC and LWP values using Eq. (1) and (2). For simplicity, we chose 100,000 CCN samples from a given CCN range and calculated the

corresponding 'true' CER and COD values. Then, random error from a normal distribution was added to the CER and COD distributions, respectively. Unphysical negative CER and COD values were removed before new corresponding CDNC and LWP values were calculated. The normal distributions are centered on zero and we quote the standard deviation of the error distribution in the text. We did this for both relative and absolute errors.

### MODIS L2 measurements

We used the level-2 (L2) Moderate Resolution Imaging Spectroradiometer (MODIS) cloud product from Collection 6.1[22] to obtain COD and CER, as well as aerosol product[23] to obtain AOD and Ångström Exponent. We intentionally selected regions of marine clouds that have been previously frequently studied and thus a reasonable choice was to select the same four areas as in Painemal et al.[24]: two regions in the Pacific Ocean (Pacific North 20–35°N, 150–110°W; Pacific South 10–30°S, 110–70°W) and two regions in the Atlantic Ocean (Atlantic North 20–35°N, 50–10°W; Atlantic South 0–30°S, 15°W–15°E). In calculating CDNC and LWP according to Eq. (1) and (2), we included only liquid, single-layer clouds with a top warmer than 268 K at 1 km resolution. We applied exactly the same filtering as suggested in Gryspeerdt et al.[10], for instance, pixels with COD < 4 or CER < 4 μm were excluded due to the increased uncertainty in retrievals for these values.

Data from years 2014–2018 was aggregated to 1 × 1° and 0.25 × 0.25° bins, respectively. For Fig. 1a the Pacific North region was divided into smaller subregions, as shown in Supplementary Fig. 1. For the daily plots we calculated the average ln(LWP) for 15 ln(CDNC) bins, with CDNC ranging from -1 to 1000 cm$^{-3}$. Each line in Fig. 1b represents CDNC-binned mean LWP values for a single day. For these cases, the daily data plots are limited to days with more than 150 samples, and the binned means are calculated only for bins with more than four samples. In Fig. 1c the data are collected from smaller areas, and the lines correspond to days with more than 100 samples, and the binned means are calculated if there are more than 3 samples in the bin. Moreover, the standard deviation of AI within the sampling area was calculated for each day respectively, and only those days were included for which this value is less than 0.04.

## Data availability

All MODIS data used in this study are open data and were obtained from the NASA Level-1 and Atmosphere Archive & Distribution System Distributed Active Archive Center (LAADS DAAC) https://ladsweb.modaps.eosdis.nasa.gov/. The simulated data generated in this study have been deposited in the Zenodo database under accession code https://doi.org/10.5281/zenodo.7100536[25].

## Code availability

The scripts used in producing the figures in this manuscript are available at https://doi.org/10.5281/zenodo.7100536[25].

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

## Acknowledgements

Funding for A.A., A.L., H.K., and J.Q. is partly provided by the FORCeS project funded by the European Union's Horizon 2020 programme, Grant Agreement No 821205. A.A. and A.L. also acknowledge the funding by Academy of Finland grant No 337549 (Atmosphere and Climate Competence Centre, ACCC). H.K. acknowledges the funding by Academy of Finland project No 317390. J.Q. acknowledges funding by the German Research Foundation (Deutsche Forschungsgemeinschaft, project No 446279238 / GZ QU 311/27-1). D.P.G. was supported by the National Environmental Research Council (NERC) national capability grant for The North Atlantic Climate System Integrated Study (ACSIS) program (grant NE/N018001/1) via NCAS and by the NERC ADVANCE Standard Grant project (NE/T006897/1). E.G. was supported by a Royal Society University Research Fellowship (URF/R1/191602)

## Author contributions

A.A. conceived the study and A.A., A.L., P.K., T.V., and H.K. designed the study question and analysis. A.A., A.L., and T.V. performed the satellite analysis and simulated datasets. A.A., A.L., P.K., T.V., N.B., D.G., E.G., J.Q., and H.K. contributed to analyzing and interpreting the results. A.A. led the manuscript writing with contribution from A.L., P.K., T.V., N.B., D.G., E.G., J.Q., and H.K.

## Competing interests

The authors declare no competing interests.
