## [Peer Review File · Nature Communications]

Aerosol effects on clouds are concealed by natural cloud heterogeneity and satellite retrieval errorsReviewer #1 (Remarks to the Author):

Review is attached.

Reviewer #1 Attachment on the following page

Review on „*Aerosol effects on clouds are concealed by natural cloud heterogeneity and satellite retrieval errors*“

by Arola et al. subm. to Nature Communications

Summary:

This paper tackles an outstanding issue in cloud physics. Namely the certainty and sign of liquid water path adjustments. Both confounding meteorological variability, spatial cloud heterogeneity and retrieval errors may impact the diagnosed relationship between cloud droplet number concentration (CDNC) and liquid water path (LWP). The paper exemplifies that the variation of spatial and temporal scales over which this relationship is diagnosed generates only marginal impacts. Meanwhile biases in cloud retrievals of cloud optical depth (COD) or cloud effective radius (REF) can change not only the magnitude of the adjustment, but the sign itself. The authors further show that this is particularly the case for biases in REF retrievals and the diagnosed CDNC (as opposed to LWP). In synthetically generated datasets the authors show that retrieval errors of 15% or more are sufficient to flip the sign of the cloud adjustment.

This study tackles an issue that remains unsolved and nicely demonstrates how important it is to stress-test and improve our methodology in providing a robust assessment of the LWP adjustment in subtropical stratocumuli. I have a couple of conceptual points, which I would like the authors to clarify prior to publication.

General remarks:

1. You attempt to remove meteorological variability by reducing either the temporal timescale over which these relationships are averaged, or the spatial scale. I am not sure that I agree with the premise that a reduction in either can really remove the influence of meteorology. An area of $5^{\circ} \times 5^{\circ}$ in the North Pacific still corresponds to an area of at least $400 \times 400 \text{ km}^2$. This incorporates the spatial scale of e.g. fronts.

I show an example of this by looking at yesterday's satellite images. This is a 5×5 deg image from your region taken on May 17th 2022:

Figure: Visible Image from worldview.earthdata.nasa.gov (May 17th 2022)

We acknowledge the use of imagery from the NASA Worldview application (<https://worldview.earthdata.nasa.gov>), part of the NASA Earth Observing System Data and Information System (EOSDIS).

Even if you decreased domain size further, I am not confident you can exclude meteorological variability fully in this manner, because the considered pixel will evolve under different meteorological conditions. I would argue that the absence of changes between the different panels of Fig. 1 is due to the fact that the compounding meteorological variability is still there.

2. It is quite striking how retrieval errors in COT and REF can propagate through to the diagnosed LWP adjustment and even change its sign. But what about the step in between? Even if you account for retrieval uncertainties of the actual retrievals, there is still the diagnosis of Nd and LWP. Here you want to assess the robustness of satellite-based estimates of the climatological Nd-LWP relationship and how it can be improved. Equation 1 for Nd contains 3 parameters which we know to be based on imperfect assumptions (especially a). How robust is the Nd-LWP relationship to variations in these parameters and isn't this potentially the larger lever? I am not suggesting that you have to necessarily do this analysis. However, if you wish not to do it, I would like to ask you to discuss different sources of uncertainty in these relationships and which ones are most likely to propagate through to LWP susceptibility.
3. My final remark regards the issue of spatial heterogeneity and the impact of filtering and relates to your statements in data selection L287-L291. My understanding is that the selection criteria imposed on the CDNC retrieval can really shrink your dataset, but of course enhance retrieval certainty in the remaining data. If you look at the points remaining in your e.g. $5^{\circ} \times 5^{\circ}$ region following the criteria selection: What is the average percentage of points you are left with as compared to all cloudy pixels where COD and REF are retrieved and thus how well do you capture spatial heterogeneity (what you refer here to as natural variability) in the remaining quality-controlled dataset? What I am trying to get at is if the sampling itself can skew the result by not capturing the natural variability of the system, since only a small sub-space remains where trust in the retrievals is high (though your results suggest that even that may not be the case).

Edits:

The manuscript is very well written, logically structured and clear. I only have minor suggested edits regarding figure labels.

Figure 1: In the top-left of the panels you give quantities that are not formally defined (e.g. N, Nday, NB). Please include a definition in the text or caption.

Figure2: include abbreviations for RE and AE in the caption for clarity.

Reviewer #2 (Remarks to the Author):

The manuscript provides an interpretation for the commonly observed negative correlation between liquid water path (LWP) and cloud droplet number concentration (CDNC). The authors argue that the anti-correlation is likely driven by the combined effect of spatial variability of clouds (which possibly violates the physical assumptions required for computing CDNC) and satellite artifacts. The authors conclude that these effects are masking the rapid adjustment of clouds associated with changes in aerosol concentration. The topic is important as the research community generally neglects biases in both satellite observations and physical assumptions when performing aerosol-cloud-interactions studies. The authors' conclusion also supports this reviewers' view on the subject, that is, the so-called "second indirect effects" are difficult, if not impossible, to observe with satellite data. This manuscript makes an important contribution by alerting the community that satellite-based assessments of aerosol-cloud interactions might be biased low, or even unphysical, especially when the sign of the LWP-CDNC slope is negative.

The exercise of creating synthetic observations with random errors is simple yet quite compelling. The conclusion that errors in cloud droplet effective radius (CER) are the single most important source of uncertainty in aerosol cloud interaction (ACI) estimates (by propagating uncertainties to CDNC) is well known. In fact, it can be easily concluded from the CDNC equation. In spite of this, the study is relevant. A general disagreement I have with the authors is that while they show plausible causes for artificial LWP-CDNC correlations, 1) the analysis cannot rule out the occurrence of physical anticorrelations between LWP and CDNC, and 2) it is unknown the magnitude of random errors in satellite data, and thus, we do not know if these errors are sufficient to explain artificial CDNC-LWP relationship when derived from satellite data. Regarding 1), one could hypothesize that droplet collision can impact the cloud microphysics by increasing the droplet size, decreasing CDNC, and enhancing LWP. I would not be surprised if satellite retrievals are able to capture these relationships, as in situ based studies have shown that MODIS retrievals show some skill in detecting precipitation. In other words, precipitation is a key mechanism missing in the manuscript, and a possible explanation for the MODIS relationship depicted in Figure 1. 2) Do we know the magnitude of satellite errors? At least, MODIS CDNC compare surprisingly well with in-situ observations in subtropical clouds. It would be interesting to make figure 1) for two regions where uncertainties in satellite data are expected to be dissimilar (e.g. homogeneous stratocumulus clouds vs shallow convective clouds and/or vs broken clouds).

Other comments

Line 31-33: Instead of "suggested", it would be more accurate to say that modeling evidence shows that entrainment can play a role

Lin 67: "MODIS retrievals" instead of "MODIS data"

Line 74: remove "...of all..."

Line 75: remove "long time" as the concept is relative

Line 116: Add commas before and after: "strictly speaking"

Line 168: I would argue that progress should be made in characterizing uncertainties in CDNC.

Reviewer #3 (Remarks to the Author):

Review of "Aerosol effects on clouds are concealed by natural cloud heterogeneity and satellite retrieval errors" by Arola et al. (MS#NCOMMS-22-14292-T)

General comments:

This study investigates causalities of increased and decreased cloud liquid water path (LWP) responses to changes in cloud droplet number concentration (CDNC) reported in several satellite-based studies. The authors employed a simple theoretical model to look

at how the errors in cloud optical depth (COD) and cloud droplet effective radius (CER) affect CDNC-LWP relationships. By perturbing both COT and CER retrievals and thus also varying LWP and CDNC, the authors showed that the assumed positive LWP adjustment can be negative through the errors. The findings will assist appropriate interpretations of satellite-derived cloud water adjustments to aerosol perturbations.

The subject matter of this work is important to reduce uncertainties in aerosol-cloud interactions, and the writing is lucid and concise. However, I have a few concerns about their analysis and discussions. The first point is that the authors use LWP and CDNC from the same retrievals through COD and CER, suggesting that the errors in COD and CER will propagate directly to LWP and CDNC, thus biases in CDNC-LWP relationships. Given that the LWP is linearly proportional to COD and CER, and that the CDNC is proportional to $COD^{0.5}$ and $CER^{-2.5}$, the CDNC-LWP relation fundamentally links to the CER-COD relation, which requires more quantitative discussions of the error propagation. Secondly, there are few discussions of uncertainties and errors in LWP adjustments due to precipitation. Satellite retrieval uses adiabatic assumption for estimating LWP and CDNC, which is no longer valid in raining clouds. Although precipitation scavenging reduces CDNC and aerosols thus causing bias in estimated LWP adjustments, the conclusion of the present paper attributes the bias only to the retrieval errors in COD and CER. The reviewer also founds that the past key studies (e.g., Chen et al., 2014; Michibata et al., 2016; Ma et al., 2018) were missed and not discussed enough even though they are closely related to the present paper. These are, however, fixable through revisions, and I feel this paper would be suitable for publication in Nature Communications. The specific issues are listed below but are almost minor presentation issues.

Specific comments:

1. Introduction (Lines 20-27; 44-48): Although ERF_{aci} for only liquid cloud is discussed here, ERF_{aci} will be essentially controlled by precipitation as well. Raindrop also weakens the magnitude of the ACI via coalescence scavenging. A more recent study reports that falling snow also effectively mitigates the aerosol-induced cloud water increases through the riming of cloud droplets on snow. It might be helpful for broad readers if the authors add discussions of how these precipitation-driven buffering link to uncertainties in estimating the cloud adjustment.
2. Line 80-86: The LWP vs CDNC relationships are also strongly affected by the occurrence of precipitation (e.g., Chen et al., 2014; McCoy et al., 2020) as it can cause retrieval errors in CDNC via adiabatic cloud assumption. I recommend the authors use other satellites as well (e.g., CloudSAT, or GPM that cover the analysis period) to distinguish whether clouds are non-raining or raining. And also, is it possible that the CDNC-LWP relationship in broken clouds might be influenced by the errors from the adiabatic cloud assumption?
3. Line 104-107: The authors argued the cause of varying CDNC-LWP relationships, but I don't think the updraft velocity, spatial variability, and cloud-top entrainment would be such a large issue in marine stratocumuli the authors analyzed. As shown in comment #2 above, discussions of how the occurrence of precipitation affects the CDNC-LWP pattern would be helpful for readers. Or, adding discussion that retrieval limitations from satellites also cause uncertainties in ERF_{aci} estimates (Ma et al., 2018) would be helpful for readers.
4. Line 168-172: Isn't this result expected from the calculations of LWP and CDNC? That is, LWP is linearly proportional to COT and CER whereas CDNC is proportional to $COT^{0.5}$ and $CER^{-2.5}$, suggesting that the CDNC-LWP relation is more sensitive to the CER retrieval. I feel that it would be better to describe the equations behind the relationship. Perhaps, the authors could add a plot that shows how the CDNC-LWP and CER-COD relationships covary (see also comment #5 below).
5. Line 173-175: Could the authors provide information about which cloud regimes are more sensitive to retrieval errors, through the combined use of the CDNC-LWP relation and CER-COD relation?
6. Line 182-184: Are there any references for these values?
7. Line 212-217: Maybe you could add a brief discussion about how such errors in COT and/or CER occur in the actual satellite retrievals.

0. Line 289: Just to check, why not 273 K for liquid and ice partitioning? How does the threshold affect the result?

1. Figure 3: There is little discussion about Figures 3 in the text. Also, could you describe why the figure uses a color bar for CCN but not CER as in Figure 2?

0. There is no explanation of extended figures 1-3, 5, and 6, but they seem to be important for the author's analyses. Could you add descriptions in the main text or extended data file?

We would like to first express our thanks to the reviewers for their especially useful and constructive comments. The point-by-point responses are below with a regular font after each reviewer's point, which in turn are in *italic font*. Moreover, the main remarks / major points of each reviewer are numbered to make them easier to find, because a few times in our response we also refer to our response we gave to another reviewer's very similar comment.

Reviewer #1 (Remarks to the author)

Summary:

This paper tackles an outstanding issue in cloud physics. Namely the certainty and sign of liquid water path adjustments. Both confounding meteorological variability, spatial cloud heterogeneity and retrieval errors may impact the diagnosed relationship between cloud droplet number concentration (CDNC) and liquid water path (LWP). The paper exemplifies that the variation of spatial and temporal scales over which this relationship is diagnosed generates only marginal impacts. Meanwhile biases in cloud retrievals of cloud optical depth (COD) or cloud effective radius (REF) can change not only the magnitude of the adjustment, but the sign itself. The authors further show that this is particularly the case for biases in REF retrievals and the diagnosed CDNC (as opposed to LWP). In synthetically generated datasets the authors show that retrieval errors of 15% or more are sufficient to flip the sign of the cloud adjustment.

This study tackles an issue that remains unsolved and nicely demonstrates how important it is to stress-test and improve our methodology in providing a robust assessment of the LWP adjustment in sub-tropical stratocumuli. I have a couple of conceptual points, which I would like the authors to clarify prior to publication.

General remarks:

1. You attempt to remove meteorological variability by reducing either the temporal timescale over which these relationships are averaged, or the spatial scale. I am not sure that I agree with the premise that a reduction in either can really remove the influence of meteorology.

An area of 5°x5° in the North Pacific still corresponds to an area of at least 400x400 km². This incorporates the spatial scale of e.g. fronts. I show an example of this by looking at yesterday's satellite images. This is a 5x5deg image from your region taken on May 17th 2022:

Figure: Visible Image from worldview.earthdata.nasa.gov (May 17th 2022)
We acknowledge the use of imagery from the NASA Worldview application (<https://worldview.earthdata.nasa.gov>), part of the NASA Earth Observing System Data and Information System (EOSDIS).

Even if you decreased domain size further, I am not confident you can exclude meteorological variability fully in this manner, because the considered pixel will evolve under different meteorological conditions. I would argue that the absence of changes between the different panels of Fig. 1 is due to the fact that the compounding meteorological variability is still there.

Response 1: We think that the meteorological variability was indeed significantly reduced by narrowing down the spatial scale, although we do agree that likely some and perhaps quite significant meteorological variability can remain within $5 \times 5^\circ$ regions. However, this fact does not reduce the strength and significance of our main arguments and results, for the reasons we explain next.

Below in Figure 1, we show cloud fields quite similar to what the reviewer chose, but in this case from south Pacific, since this image was readily available from our previous analyses and nicely serves the purpose here as well. The lower panels show COT and CER and upper panels LWP and CDNC, calculated using this information of cloud optical thickness and cloud effective radius at 1km L2 MODIS data. Figure 2, on the other hand, shows the results when both CDNC and LWP are aggregated to $1^\circ \times 1^\circ$ using L2 1km COD and CER data, as is typically done in the literature. There is significant spatial variability in COT and CER, and thus in CDNC and LWP, even if trying to exclude any CCN variability (through Aerosol Index, which was the proxy for CCN). And as is illustrated by the arrows in Figure 2, if broadly looking at the cloud fields, it seems that LWP is mostly decreasing when CDNC is increasing. This spatial behavior naturally results from the spatial variation of COT and CER, and the reasons for that stem from different spatial scales; from the large-scale meteorology, from the small-scale cloud variability (for instance, due to the true variations in the adiabatic fraction), and, of course, from the spatial aerosol variability. Our main argument is that the potential impact from both from the small-scale cloud variability (natural cloud heterogeneity) and retrieval errors have not been sufficiently considered and has been biasing the LWP vs. CDNC interpretation in many previous studies. And as can be seen, while this biasing effect is there in the coarse resolution, it exists there in the finest resolution as well. This brings a more general consideration that large-scale meteorological variability and small-scale cloud

variability manifest themselves quite similarly in this kind of LWP adjustment analysis. Therefore, when the data are gathered as has been usually done for

LWP and CDNC analysis, this kind of cloud heterogeneity, if not accounted for, is biasing the results through an effect that is not related to the cloud response to aerosol change.

In light of the above comments, we have reduced the strength of the claim to have removed meteorological variability by modifying the following sentence from:

"In such cases any relationships between LWP and CDNC should be due to factors other than confounding meteorological factors or changes in CCN concentrations."

To: "In such cases the dependence of relationships between LWP and CDNC on confounding large-scale meteorological factors or changes in CCN concentrations should be reduced."

Figure R1. Liquid Water Path (LWP), Cloud Droplet Number concentration (CDNC), Cloud Optical Thickness (COT) and Cloud Droplet Effective Radius (CER) fields over ocean close to the coast of Chile. COT and CER are from MODIS L2 at 1km resolution and CDNC and LWP are calculated based on the Equations 1 and 2 of the manuscript.

We acknowledge the use of imagery from the NASA Worldview application (<https://worldview.earthdata.nasa.gov>), part of the NASA Earth Observing System Data and Information System (EOSDIS).

Figure R2. Same data as in the Figure R1, but now aggregated to 1x1° resolution.

2. It is quite striking how retrieval errors in COT and REF can propagate through to the diagnosed LWP adjustment and even change its sign. But what about the step in between? Even if you account for retrieval uncertainties of the actual retrievals, there is still the diagnosis of Nd and LWP. Here you want to assess the robustness of satellite-based estimates of the climatological Nd-LWP relationship and how it can be improved. Equation 1 for Nd contains 3 parameters which we know to be based on imperfect assumptions (especially a). How robust is the Nd-LWP relationship to variations in these parameters and isn't this potentially the larger lever? I am not suggesting that you have to necessarily do this analysis. However, if you wish not to do it, I would like to ask you to discuss different sources of uncertainty in these relationships and which ones are most likely to propagate through to LWP susceptibility.

Response2: The equation for CDNC calculation is: $CDNC = \alpha * COT^{0.5} / CER^{2.5}$. We have introduced various levels of variability/error into COT and CER, while the equation shows that the variability in alpha (for instance because the adiabatic fraction deviates from 0.8, which is assumed in the bulk coefficient of alpha) introduces similar effect to COT but more strongly (power of 1 compared to the power of 0.5 for COT). Below is a figure of that

additional sensitivity study, where COT and CER error was set to 15%, but additionally alpha had various levels of error/variability. These new results suggest that the descending branch of LWP vs. CDNC likely starts a bit later as alpha variability increases, but if the alpha variability is a random variability/error, our main finding of a biasing impact from COT and CER variability/error is still strongly present even with a 25% alpha error. We have discussed this in the revised version (see tracked changed version of the revised manuscript).

Figure R3. Median values of LWP as a function of CDNC, resulting from varying levels of relative error in α -term in the Equation 1 of the manuscript. Errors of both COT and CER were 15% in this simulation.

3. My final remark regards the issue of spatial heterogeneity and the impact of filtering and relates to your statements in data selection L287-L291. My understanding is that the selection criteria imposed on the CDNC retrieval can really shrink your dataset, but of course enhance retrieval certainty in the remaining data. If you look at the points remaining in your e.g. 5°x5° region following the criteria selection: What is the average percentage of points you are left with as compared to all cloudy pixels where COD and REF are retrieved and thus how well do you capture spatial heterogeneity (what you refer here to as natural variability) in the remaining quality-controlled dataset? What I am trying to get at is if the sampling itself can skew the result by not capturing the natural variability of the system, since only a small sub-space remains where trust in the retrievals is high (though your results suggest that even that may not be the case).

Response 3: This is an interesting point indeed. And to continue on the subject, besides what fraction of data remains, it is especially important and interesting to see whether those data cover rather extensive and entire cloud covered regions. Because, as the reviewer stated, the essential thing is to make sure that we actually capture spatial heterogeneities, which we argue cause bias in LWP vs. CDNC relationship. Below are the

data fractions remaining after applying all the filters detailed in Gryspeerdt et al. (2019) if compared to all available cloud retrievals: "Filtered data"/"All data"*100%.

AtlanticN 20.4%
AtlanticS 35.8%
PacificN 29.4%
PacificS 38.8%

As can be seen, quite large amount of data "survives" the filters. And more importantly, extensive entire cloud areas remain in the filtered data. A typical example is shown below. The upper panel shows the visible image, middle panel "All data" and lower panel "Filtered data". Point to note: there are sensor and solar angle limits also, thus only part of the swath is included in the lowest panel of filtered data.

Figure R4. Visible cloud field from Pacific South region (upper panel), all pixels where the COT and CER retrieval was done are shown in red (middle panel). In the lowest panel are those cloud retrieval pixels that are left after the filtering was applied (following Gryspeerdt et al. 2019).

We acknowledge the use of imagery from the NASA Worldview application (<https://worldview.earthdata.nasa.gov>), part of the NASA Earth Observing System Data and Information System (EOSDIS).

Edits:

The manuscript is very well written, logically structured and clear. I only have minor suggested edits regarding figure labels.

Figure 1: In the top-left of the panels you give quantities that are not formally defined (e.g. N , N_{day} , NB). Please include a definition in the text or caption.

Response: These are now explained in the figure caption.

Figure2: include abbreviations for RE and AE in the caption for clarity.

Response: Abbreviations now included in the figure caption.

Reviewer #2 (Remarks to the Author):

The manuscript provides an interpretation for the commonly observed negative correlation between liquid water path (LWP) and cloud droplet number concentration (CDNC). The authors argue that the anti-correlation is likely driven by the combined effect of spatial variability of clouds (which possibly violates the physical assumptions required for computing CDNC) and satellite artifacts. The authors conclude that these effects are masking the rapid adjustment of clouds associated with changes in aerosol concentration. The topic is important as the research community generally neglects biases in both satellite observations and physical assumptions when performing aerosol-cloud-interactions studies. The authors' conclusion also supports this reviewers' view on the subject, that is, the so-called "second indirect effects" are difficult, if not impossible, to observe with satellite data. This manuscript makes an important contribution by alerting the community that satellite-based assessments of aerosol-cloud interactions might be biased low, or even unphysical, especially when the sign of the LWP-CDNC slope is negative.

The exercise of creating synthetic observations with random errors is simple yet quite compelling. The conclusion that errors in cloud droplet effective radius (CER) are the single most important source of uncertainty in aerosol cloud interaction (ACI) estimates (by propagating uncertainties to CDNC) is well known. In fact, it can be easily concluded from the CDNC equation. In spite of this, the study is relevant. A general disagreement I have with the authors is that while they show plausible causes for artificial LWP-CDNC correlations, 1) the analysis cannot rule out the occurrence of physical anticorrelations between LWP and CDNC,

Response 1: Our aim was to show that the spatial cloud heterogeneity (or retrieval errors of COT and CER) is causing a negative bias to the $d\ln LWP/d\ln CDNC$ sensitivity, however we did not argue that there could not be any true physical anticorrelations between LWP and CDNC that would then result in negative value for LWP adjustment. And we discussed and mentioned some possible physical reasons in our manuscript. But our main message was, as the reviewer formulated it above, that "these effects are masking the rapid adjustment of clouds associated with changes in aerosol concentration". For instance, if the true $d\ln LWP/d\ln CDNC$ sensitivity was -0.1, also in this case the estimated sensitivity would become more negative, if the spatial variability (or retrieval errors) is not considered in the analysis. Related to this point and to make this even clearer, we also modified in the revised manuscript the following sentence:

"and they are indeed entirely plausible explanations that have likely played some role in those analyzed cases"

to "and there are indeed entirely plausible physical mechanisms that could have produced the inferred relationships".

and 2) it is unknown the magnitude of random errors in satellite data, and thus, we do not know if these errors are sufficient to explain artificial CDNC-LWP relationship when derived from satellite data.

Response 2: In the revised version we discuss more about the influence of COT and CER uncertainties. However, we would like to stress that the spatial variability of (even if error-free) COT and CER is likely playing an equally important role in this biasing effect than possible retrieval errors, as is also illustrated by the cloud fields of the Figures R1 and R2 above. However, it is to be stressed that the errors we applied in our simulations were also likely very plausible. For instance, in Grosvenor et al. 2018, the following statement was given to assess the level of error in CER: "Due to resolved and unresolved heterogeneity, an uncertainty in r_e of 17% was assessed in section 2.4.3 and that due to instrument uncertainty was estimated as 10% (section 2.4.7) giving an overall error of 27%."

Regarding 1), one could hypothesize that droplet collision can impact the cloud microphysics by increasing the droplet size, decreasing CDNC, and enhancing LWP. I would not be surprised if satellite retrievals are able to capture these relationships, as in situ based studies have shown that MODIS retrievals show some skill in detecting precipitation. In other words, precipitation is a key mechanism missing in the manuscript, and a possible explanation for the MODIS relationship depicted in Figure 1.

Response 3: Reviewer #3 also brought up the importance of distinguishing raining and non-raining clouds in this kind of analysis. We do agree that it is an important aspect to consider, however we argue it cannot play a significant role in our main finding, for the following reasons.

Our main finding and argument were that when CDNC increases sufficiently, LWP starts decreasing due to the biasing effect arising from COT and CER spatial variability. It follows that the behavior is at its strongest when CER values are small. On the other hand, one approach to separate raining and non-raining clouds has been to apply a CER threshold (typically of $15\mu\text{m}$), so that the cases with CER less than that the threshold are non-raining clouds. This means that when using this kind of separation, the biasing effect most strongly influences in the conditions of non-raining clouds.

We made some additional analysis by separating our data set to include only cases when CER was smaller than $15\mu\text{m}$. In the following figure, several cases are included. In the LHS plot all data are included, but additionally both cases of $\text{CER}=30\text{mm}$ and $\text{CER}=15\mu\text{m}$ are shown by black and dashed red line, respectively. In the middle panel, such a subset of all data is shown when CER was less than $15\mu\text{m}$. Then, in the RHS plot even a more stringent $\text{CER}=15\mu\text{m}$ threshold was applied as follows: if there was any case of $\text{CER}>15\mu\text{m}$ in the entire Pacific North area, then the full day was entirely excluded. Regardless of how we separately focus on non-raining clouds, it is apparent that the biasing effect becomes most obvious actually on non-raining clouds, as explained above. In every case, there is an initial increase in LWP as CDNC increases, but as can be concluded from the figures it does follow the threshold and is artificially influenced by the strict limit in CER (as illustrated by the diagonal black and red-dashed lines).

Figure R5. In the left-hand-side plot all data are included with the effective radius of CER=30µm and CER=15µm shown by black and dashed red line, respectively. In the middle panel, subset with CER less than 15µm are included. In the right-hand-side plot a more stringent CER=15µm threshold was applied as follows: if there was any case of CER larger than 15µm in the entire Pacific North area in a given day, then the full day was entirely excluded.

2) Do we know the magnitude of satellite errors? At least, MODIS CDNC compare surprisingly well with in-situ observations in subtropical clouds. It would be interesting to make figure 1) for two regions where uncertainties in satellite data are expected to be dissimilar (e.g. homogeneous stratocumulus clouds vs shallow convective clouds and/or vs broken clouds).

Response 4: Pixel-level uncertainties of MODIS COT and CER are available for 1km L2 product, and we have now further utilized them (see below). However, we want to first emphasize that the meaning and interpretation of these uncertainties becomes unclear with aggregated data at 1x1° resolution. Secondly, we also want to emphasize is that we do not assume that the level of COT and CER retrieval uncertainty should be significant enough to explain the biasing pattern in LWP vs. CDNC that have been observed; since the natural heterogeneity of cloud fields and thus the variability in COT and CER, unrelated to aerosol amounts, most probably plays an equally important role. And the third point to mention, related to the comment above: even if the spatially or temporally averaged CDNC might compare quite well with the in-situ measurement, it does not mean that the spatial variation of these fields cannot be biasing the satellite-based $d\ln LWP/d\ln CDNC$. Indeed, it has been found that the Nd (e.g. Gryspeerd et al, 2022) and LWP (e.g. Seethala and Horvath, 2010) retrievals appear reasonable on average, which suggests that errors in the formulation of the equations are likely not particularly noticeable, but when correlated errors when LWP vs. CDNC are analyzed, they become important as we have demonstrated in our paper.

In the following figure we show the number of 1x1° aggregated pixels in two cases: "all data" in the

upper panel and cases when only 1km MODIS cloud retrievals of COD and CER are aggregated if the uncertainty was less than 10% (middle panel). Lower panel in turn shows the difference in the data amount ($N_{original} - N_{filtered}$). Overall and quite understandably the COT and CER uncertainties decrease when their absolute values increase, thus the number of "less uncertain" pixels reduce towards land where CER values are smaller.

We think that a comprehensive analysis by defining different cloud regimes, for instance based on COT and cloud top height (like in Oreopoulos et al. 2014, for instance) and then comparing typical uncertainties of COT and CER would need a separate study. Our region in Figure R6 is wide enough to give an idea about impact of uncertainty in the data sampling. And more importantly, in the following Figure R7, we see a very similar LWP vs. CDNC patterns for all sub-regions, where the overall uncertainties in CER and COT (and thus the amount of data with better accuracy) can be somewhat different. This suggests that the biasing effect remains, even when attempt was made to include only cases of COT and CER with improved retrieval accuracy; supporting the assumption that the biasing effect is more due to the natural cloud heterogeneity than retrieval uncertainties themselves (although both would produce a similar effect)

We also selected a two-month period of MODIS data from South Pacific Painemal region, so that it was far enough from the coast (thus far from strong gradients in CCN) and where often extensive and uniform cloud fields are formed. This selection was based on first spending many hours just by looking at cloud fields from NASA Worldview. Then two-month period was selected, corresponding COT and CER data gathered, Gryspeerdt et al. 2019 filtering applied and additionally restricting by pixel-level uncertainties to have a subset where COT and CER uncertainty was always below threshold of 7.5% uncertainty. In the Supplementary material one very typical case is included with short discussion in revised manuscript. And indeed, there was surprisingly little variability from day-to-day, regarding the LWP vs. CDNC patterns between the "full data" and the "least uncertain data". What one can see, and what was clear and obvious when going through the entire data set, is that the negative pattern of LWP vs. CDNC is only slightly reduced when only the least uncertain data are included. It is difficult to fully quantify how much these patterns are due to the COT and CER uncertainties and how much due to the "natural cloud heterogeneity", but this was an attempt to demonstrate that certainly both play significant roles.

Figure R6. Number of $1 \times 1^\circ$ pixels in Pacific North without applying any "uncertainty limit" (upper panel). Number of data when only those pixels are included when both CER and COT uncertainties were less than 10% (middle panel). Difference in the amount of data with and without uncertainty limit (lower panel).

Figure R7. LWP vs. CDNC over different regions (then also over regions of different overall uncertainties in COT and CER, as illustrated in R6). Solid black line shows the median of all the data and is thus same in all three panels.

Other comments

Line 31-33: Instead of "suggested", it would be more accurate to say that modeling evidence shows that entrainment can play a role

Changed as suggested by the reviewer.

Lin 67: "MODIS retrievals" instead of "MODIS data"

In our opinion word "data" suits better in this context.

Line 74: remove "...of all..."

Changed as suggested by the reviewer.

Line 75: remove "long time" as the concept is relative

Changed as suggested by the reviewer.

Line 116: Add commas before and after: "strictly speaking"

Changed as suggested by the reviewer.

Line 168: I would argue that progress should be made in characterizing uncertainties in CDNC.

Changed as follows: "This suggests that the future progress should focus on characterizing uncertainties and improving the accuracy of satellite-based CDNC estimations; ..."

Reviewer #3 (Remarks to the Author):

Review of "Aerosol effects on clouds are concealed by natural cloud heterogeneity and satellite retrieval errors" by Arola et al. (MS#NCOMMS-22-14292-T)

General comments:

This study investigates causalities of increased and decreased cloud liquid water path (LWP) responses to changes in cloud droplet number concentration (CDNC) reported in several satellite-based studies. The authors employed a simple theoretical model to look at how the errors in cloud optical depth (COD) and cloud droplet effective radius (CER) affect CDNC-LWP relationships. By perturbing both COT and CER retrievals and thus also varying LWP and CDNC, the authors showed that the assumed positive LWP adjustment can be negative through the errors. The findings will assist appropriate interpretations of satellite-derived cloud water adjustments to aerosol perturbations.

The subject matter of this work is important to reduce uncertainties in aerosol-cloud interactions, and the writing is lucid and concise. However, I have a few concerns about their analysis and discussions. The first point is that the authors use LWP and CDNC from the same retrievals through COD and CER, suggesting that the errors in COD and CER will propagate directly to LWP and CDNC, thus biases in CDNC-LWP relationships. Given that the LWP is linearly proportional to COD and CER, and that the CDNC is proportional to $COD^{0.5}$ and $CER^{-2.5}$, the CDNC-LWP relation fundamentally links to the CER-COD relation, which requires more quantitative discussions of the error propagation.

Response 1: We have added a new figure (in the Supplementary material) with a brief discussion in the main text, which hopefully better illustrates how CDNC-LWP relationship is linked to the CER-COD relation. It is also added here below. Here in addition, there are two completely random cases and for illustrative purposes only; for instance, the red line illustrates the case when LWP increases as CDNC increases - in that case CER naturally decreases, but COT would have to increase sufficiently. CER is illustrated by color and COT by white isolines. We discuss this figure briefly in the revised version, as suggested by the reviewer.

Figure R8. CDNC-LWP relationships as a function of CER and COD, calculated by using Equations 1 and 2 in the manuscript. Red and blue circles can be taken to illustrate the mean CDNC and LWP of less and more polluted case of Figure 4 in the main text. Black circle shows the corresponding change in CDNC if the LWP is constant.

Secondly, there are few discussions of uncertainties and errors in LWP adjustments due to precipitation. Satellite retrieval uses adiabatic assumption for estimating LWP and CDNC, which is no longer valid in raining clouds. Although precipitation scavenging reduces CDNC and aerosols thus causing bias in estimated LWP adjustments, the conclusion of the present paper attributes the bias only to the retrieval errors in COD and CER. The reviewer also finds that the past key studies (e.g., Chen et al., 2014; Michibata et al., 2016; Ma et al., 2018) were missed and not discussed enough even though they are closely related to the present paper. These are, however, fixable through revisions, and I feel this paper would be suitable for publication in Nature Communications. The specific issues are listed below but are almost minor presentation issues.

Response 2: The role of precipitation is discussed in the revised manuscript, as explained in more detail below. In addition, key past studies that were missed in the original manuscript are now mentioned and discussed.

Specific comments:

1. Introduction (Lines 20-27; 44-48): Although ERF_{aci} for only liquid cloud is discussed here, ERF_{aci} will be essentially controlled by precipitation as well. Raindrop also weakens the magnitude of the ACI via coalescence scavenging. A more recent study reports that falling snow also effectively mitigates the aerosol-induced cloud water increases through the riming of cloud droplets on snow. It might be helpful for broad readers if the authors add discussions of how these precipitation-driven buffering link to uncertainties in estimating the cloud adjustment.

Response 3: The importance to addressing and discussing the potential role of precipitation in satellite-based LWP adjustment studies is now discussed in the revised manuscript. Also, see our Response 3 to the Reviewer #2.

2. Line 80-86: The LWP vs CDNC relationships are also strongly affected by the occurrence of precipitation (e.g., Chen et al., 2014; McCoy et al., 2020) as it can cause retrieval errors in CDNC via adiabatic cloud assumption. I recommend the authors use other satellites as well (e.g., CloudSAT, or GPM that cover the analysis period) to distinguish whether clouds are non-raining or raining. And also, is it possible that the CDNC-LWP relationship in broken clouds might be influenced by the errors from the adiabatic cloud assumption?

Response 4: We think that the use of other satellite data would be a topic of a separate study, because such an analysis should not affect our main message in this paper about the biasing effect in the LWP adjustment, which is at its strongest in the conditions of non-raining clouds. However, we made some additional work to separate raining and non-

raining clouds and also added a brief explanation to the revised manuscript. See our Response 3 to the Reviewer #2.

0. Line 104-107: The authors argued the cause of varying CDNC-LWP relationships, but I don't think the updraft velocity, spatial variability, and cloud-top entrainment would be such a large issue in marine stratocumuli the authors analyzed. As shown in comment #2 above, discussions of how the occurrence of precipitation affects the CDNC-LWP pattern would be helpful for readers. Or, adding discussion that retrieval limitations from satellites also cause uncertainties in ERFaci estimates (Ma et al., 2018) would be helpful for readers.

Response 5: We agree, regarding the updraft velocity and other factors too, that the temporally or spatially averaged updraft velocity might not vary substantially in these conditions. However, what matters is the regional variation of updraft velocities, which we claim varies enough even in stratocumulus conditions to cause sufficient cloud heterogeneity (see for instance Figures R1 and R2 above), which then masks the true LWP adjustment.

1. Line 168-172: Isn't this result expected from the calculations of LWP and CDNC? That is, LWP is linearly proportional to COT and CER whereas CDNC is proportional to $COT^{0.5}$ and $CER^{-2.5}$, suggesting that the CDNC-LWP relation is more sensitive to the CER retrieval. I feel that it would be better to describe the equations behind the relationship. Perhaps, the authors could add a plot that shows how the CDNC-LWP and CER-COD relationships covary (see also comment #5 below).

Response 6: We believe that our new figure, shown above as a Figure R8 and added to the Supplementary material as well, is the kind of plot that the reviewer is suggesting. The related discussion in the revised version can be found from tracked changed version of the revised manuscript.

2. Line 173-175: Could the authors provide information about which cloud regimes are more sensitive to retrieval errors, through the combined use of the CDNC-LWP relation and CER-COD relation?

Response 7: We made some further analysis by considering the retrieval uncertainties, but since this point is very close to the point discussed above and raised by the Reviewer #2, we hope we sufficiently and adequately answered in the Response 4 to the Reviewer #2.

3. Line 182-184: Are there any references for these values?

Response: These values were assessed from a Table in Toll et al. 2019, as typical/reasonable values. This is now mentioned in the revised manuscript.

4. Line 212-217: Maybe you could add a brief discussion about how such errors in COT and/or CER occur in the actual satellite retrievals.

Response: Sources of COT and CER retrieval uncertainties are now briefly discussed (but we mentioned that the interested reader is recommended to read the lengthy Chapter 2.4 of Grosvenor et al. 2018 for further details and information). The reviewer referred to the "Discussion" section of our manuscript with his/her comment, while we thought it more appropriate place to add that short discussion already earlier and we added it into the last paragraph of introduction.

2. Line 289: Just to check, why not 273 K for liquid and ice partitioning? How does the threshold affect the result?

Response: We followed strictly the filtering "recipe" of Gryspeerdt et al. 2019, who used 268K. This value arises because cloud top phase is primarily determined through the cloud optical properties retrieval. As the proportion of ice clouds at -5 is very small (e.g., Kanitz et al., 2014), the threshold of 268 K increases the population of retrieved clouds globally. However, the threshold change to 273 K would likely have negligible impact in our analysis.

3. Figure 3: There is little discussion about Figures 3 in the text. Also, could you describe why the figure uses a color bar for CCN but not CER as in Figure 2?

Response: In the revised version, we changed the color bar for CER also in Figure 2.

1. There is no explanation of extended figures 1-3, 5, and 6, but they seem to be important for the author's analyses. Could you add descriptions in the main text or extended data file?

Response: Some of the figures, the reviewer referred to, were indeed not discussed. We excluded S1 and S2 (of the original Supplementary material) and now discussed all the figures in the Supplementary material, at least briefly.

REFERENCES

Gryspeerdt, E., Goren, T., Sourdeval, O., Quaas, J., Mulmenstadt, J., Dipu, S., ... & Christensen, M. (2019). Constraining the aerosol influence on cloud liquid water path. *Atmospheric Chemistry and Physics*, 19(8), 5331-5347.

Gryspeerdt, E., McCoy, D. T., Crosbie, E., Moore, R. H., Nott, G. J., Painemal, D., ... & Ziemba, L. (2022). The impact of sampling strategy on the cloud droplet number concentration estimated from satellite data. *Atmospheric Measurement Techniques*, 15(12), 3875-3892.

Seethala, C., & Horvath, A. (2010). Global assessment of AMSR-E and MODIS cloud liquid water path retrievals in warm oceanic clouds. *Journal of Geophysical Research: Atmospheres*, 115(D13).

Reviewer #1 (Remarks to the Author):

I would like to thank the authors for their detailed responses and additional analyses to address the reviewers queries. I agree with the changes made within the manuscript and fully support publication.

One minor text edit:

P3L92: "with with"

Reviewer #3 (Remarks to the Author):

2nd round review of "Aerosol effects on clouds are concealed by natural cloud heterogeneity and satellite retrieval errors" by Arola et al. (MS#NCOMMS-22-14292A)

Summary and Recommendation:

The authors have addressed my concerns. I think this paper is suitable for publication in its current form.

We would like to first express our thanks to the reviewers for their especially useful and constructive comments. The point-by-point responses are below with a regular font after each reviewer's point, which in turn are in *Italic font*.

Reviewer #1 (Remarks to the Author):

I would like to thank the authors for their detailed responses and additional analyses to address the reviewers queries. I agree with the changes made within the manuscript and fully support publication.

One minor text edit:

P3L92: "with with"

RESPONSE: This has been corrected in the revised manuscript.

Reviewer #3 (Remarks to the Author):

2nd round review of "Aerosol effects on clouds are concealed by natural cloud heterogeneity and satellite retrieval errors" by Arola et al. (MS#NCOMMS-22-14292A)

Summary and Recommendation:

The authors have addressed my concerns. I think this paper is suitable for publication in its current form.